# Boosting 3D Neuron Segmentation with 2D Vision Transformer Pre-trained on Natural Images

**Yik San Cheng**[1]                                     YCHE6976@UNI.SYDNEY.EDU.AU

**Runkai Zhao**[1]                                       RZHA9419@UNI.SYDNEY.EDU.AU

**Heng Wang**[1,2]                                       HENG.WANG.CV@GMAIL.COM

**Hanchuan Peng**[3]                                     H@BRAINTELL.ORG

**Weidong Cai**[1]                                       TOM.CAI@SYDNEY.EDU.AU

[1] *School of Computer Science, The University of Sydney, Sydney, Australia*

[2] *Data61, CSIRO, Sydney, Australia*

[3] *SEU-ALLEN Joint Center, Institute for Brain and Intelligence, Southeast University, Nanjing, China*

## Abstract

Neuron reconstruction, one of the fundamental tasks in neuroscience, rebuilds neuronal morphology from 3D light microscope imaging data. It plays a critical role in analyzing the structure-function relationship of neurons in the nervous system. However, due to the scarcity of neuron datasets and high-quality SWC annotations, it is still challenging to develop robust segmentation methods for single neuron reconstruction. To address this limitation, we aim to distill the consensus knowledge from massive natural image data to aid the segmentation model in learning the complex neuron structures. Specifically, in this work, we propose a novel training paradigm that leverages a 2D Vision Transformer model pre-trained on large-scale natural images to initialize our Transformer-based 3D neuron segmentation model with a tailored 2D-to-3D weight transferring strategy. Our method builds a knowledge sharing connection between the abundant natural and the scarce neuron image domains to improve the 3D neuron segmentation ability in a data-efficiency manner. Evaluated on a popular benchmark, BigNeuron, our method enhances neuron segmentation performance by 8.71% over the model trained from scratch with the same amount of training samples.

**Keywords:** 3D Neuron Reconstruction, Volumetric Image Segmentation, Transfer Learning, Vision Transformer, 3D Microscope Image

## 1. Introduction

Single neuron reconstruction aims to extract and digitalize tree-shaped neuron structures from 3D light microscopic images. It is essential to analyze and understand the connectivity and functionality of different neurons within the nervous system (Zhang et al., 2018; Liu et al., 2024; Gao et al., 2023; Manubens-Gil et al., 2023; Peng et al., 2021; Qiu et al., 2024). Traditional neuron reconstruction methods depend on manual labeling and hand-crafted tracing algorithms, which are labor-intensive and time-consuming (Wang et al., 2021a). Recently, learning-based techniques have been developed to extract neuron foreground pixels in a data-driven manner, including multi-scale kernel fusion (Wang et al., 2019a), atrous spatial pyramid pooling (Li and Shen, 2019), global graph reasoning (Wang et al., 2021b), cross-volume representation learning (Wang et al., 2021c), homogenous model knowledge

transfer (Wang et al., 2019b) and 3D point geometry learning (Zhao et al., 2023). Although these works prove that learning-based segmentation models can adaptively learn the complex neuron morphology in an end-to-end fashion, the limited availability of volumetric neuron image datasets with high-quality SWC annotations restricts the segmentation performance. To alleviate this problem, we propose to fully leverage the vast repository of natural image data. By distilling the consensus knowledge from these data, we strive to boost the segmentation capability in accurately interpreting neuronal morphology. Previous studies (Jiang et al., 2018; Raj et al., 2021; McBee et al., 2023) in the medical domain have demonstrated that transferring knowledge from 2D natural image data improves segmentation performance. However, utilizing 2D natural knowledge to enhance 3D neuron segmentation models remains unexplored due to the dimension and domain disparity. In this study, we design a novel training paradigm to apply a tailored 2D-to-3D weight transferring strategy on the initialization of Transformer-based 3D neuron segmentation models through the derived knowledge from 2D natural images.

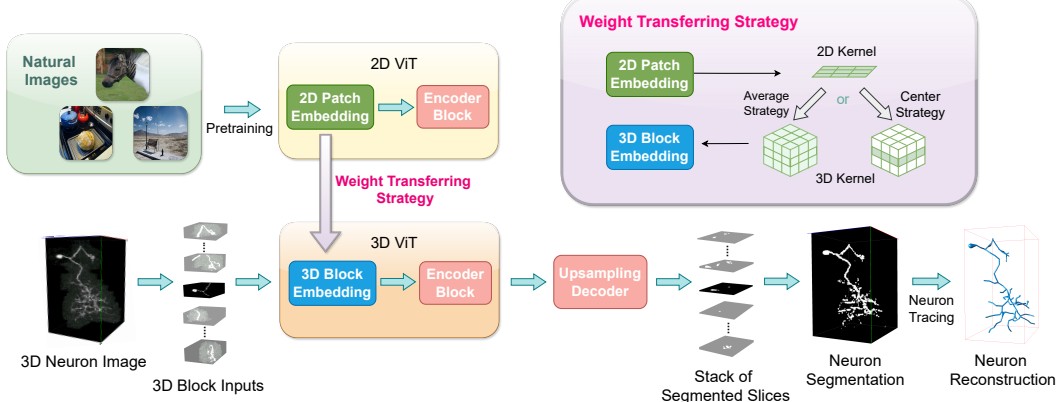

Figure 1: The overview of our proposed training paradigm for 3D neuron segmentation. The network follows an encoder-decoder structure. The 3D neuron image is first divided into several 3D blocks which are then fed to a 3D Vision Transformer (ViT) for slice segmentation. During training phase, the pre-trained weights from a 2D ViT are used to initialize the 3D ViT through a weight transferring strategy. In the end, the segmented slices are stacked together to form the final segmentation prediction which is then forwarded to produce the target SWC file through a neuron tracing method.

## 2. Methodologies

### 2.1. Dataset and Preprocessing

The neuron datatset utilized in our study is obtained from a publicly accessible Janelia dataset developed by the BigNeuron project (Peng et al., 2015; Manubens-Gil et al., 2023). In our study, this dataset is split into training, validation, and testing sets, accounting for 38, 3, and 4 of images respectively. Due to the memory limitation, we divide neuron image volumes into multiple 3D blocks with the size of $100 \times 100 \times 5$ and train only with blocks containing salient neuronal voxels (over a predefined foreground ratio). Following

the current method (Wang et al., 2019a), we apply the scale-space distance transformation technique to produce neuron segmentation ground truth.

## 2.2. Network Architecture and Weight Transferring Strategy

Our network architecture is shown in Figure 1. We leverage the consensus knowledge from DINO (Caron et al., 2021), which is a self-supervised 2D Vision Transformer (ViT) model trained on a range of large 2D natural image datasets including ImageNet and COCO. To effectively transfer the prior knowledge grabbed from 2D natural image data to our 3D neuron segmentation model, we propose to apply a weight transferring strategy (Zhang et al., 2022) to facilitate the training of our 3D neuron ViT. Specifically, we investigate two variants as shown in Figure 1. For the average strategy, we treat all the slices in the block input equally. We duplicate the pre-trained weights of 2D kernels in the 2D patch embedding layer by the number of block depth to create the 3D kernels which are then transferred to the 3D block embedding layer after being divided by the number of the duplication. For the center strategy, we only transfer the pre-trained weights in the 2D kernels to the center slice of the 3D kernels but make neighboring slices to zero.

Table 1: Neuron Segmentation Performance.

| Model | Pre-trained Weights | Transferring Strategy | Input Depth | Mean Dice↑ | Mean Hd95↓ |
|---|---|---|---|---|---|
| 2D ViT | / | / | 1 | 0.4469 | 7.649 |
| | ✓ | / | 1 | **0.4876** | **2.893** |
| 3D ViT | / | / | 5 | 0.4174 | 37.943 |
| | ✓ | Average | 5 | 0.4942 | 2.863 |
| | ✓ | Center | 5 | **0.5045** | **2.810** |

## 3. Results and Conclusion

As presented in Table 1, leveraging the knowledge extracted from 2D natural images greatly improves the neuron segmentation performance compared with models trained with random initialization, from 0.4174 to 0.5045 in mean Dice and 37.943 to 2.810 in mean 95% Hausdorff distance (Hd95). It is also found that the center weight transferring strategy achieves the best performance. In addition, including depth information can outperform the slice-to-slice based 2D ViT method.

In conclusion, our study demonstrates the effectiveness of transferring 2D prior knowledge from natural image data to improve 3D neuron segmentation without introducing additional overhead, which mitigates the data scarcity problem in neuron study. With the same amount of training samples, our 2D-to-3D weight transferring training paradigm can boost the neuron segmentation performance by 8.71% on the BigNeuron benchmark.

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
