# OpenReview forum: "Boosting 3D Neuron Segmentation with 2D Vision Transformer Pre-trained on Natural Images"
_MIDL.io/2024/Short_Papers — MIDL 2024 Short Papers_

### Official Review · Reviewer_wrG6 · 2024-04-25

**Confidence:** 5
**Final Rating:** 4

**Review:**

This paper presents a strategy that leverages pre-trained vision transformer weights from 2D natural images to benefit 3D transformer-based neuron reconstruction/segmentation. Experimental results demonstrate that the proposed method outperforms 2D/3D vision transformers without utilizing pre-trained network weights from 2D natural images.

Pros:

The methodology is straightforward, making it easy for readers to understand.

Cons:

-It would be beneficial to include results comparing the proposed method with a 2D vision transformer pre-trained specifically on 2D neuron segmentation tasks.

-This is minor, but terminologies such as "SWC" should be clearly defined for readers' understanding.

Overall, the paper provides an interesting finding for the MIDL community.

---

### Decision · Program_Chairs · 2024-04-26

Accept